# Blunted Cortisol Awakening Response Is Associated with External Attribution Bias Among Individuals with Personality Disorders

**DOI:** 10.3390/brainsci14101040

**Published:** 2024-10-20

**Authors:** Bogna Bogudzińska, Julian Maciaszek, Bartłomiej Stańczykiewicz, Tomasz Bielawski, Agnieszka Dybek, Julia Alejnikowa, Tomasz Pawłowski, Błażej Misiak

**Affiliations:** 1Department of Psychiatry, Wroclaw Medical University, 50-367 Wroclaw, Poland; julian.maciaszek@umw.edu.pl (J.M.); tomasz.bielawski@umw.edu.pl (T.B.); agdybek@gmail.com (A.D.); akiira591@gmail.com (J.A.); blazej.misiak@umw.edu.pl (B.M.); 2Division of Consultation Psychiatry and Neuroscience, Department of Psychiatry, Wroclaw Medical University, 50-367 Wroclaw, Poland; bartlomiej.stanczykiewicz@umw.edu.pl; 3Division of Psychotherapy and Psychosomatic Disorders, Department of Psychiatry, Wroclaw Medical University, 50-367 Wroclaw, Poland; tomasz.pawlowski@umw.edu.pl

**Keywords:** cortisol, personality disorder, stress, cognitive bias

## Abstract

Background/Objectives: The dysregulation of the hypothalamic–pituitary–adrenal (HPA) axis has been associated with various mental disorders. One of the most commonly described parameters of HPA axis functioning is the cortisol awakening response (CAR). To date, few studies have been conducted on the relationship between personality disorders and CAR. The present study aimed to compare the CAR between individuals with personality disorders and healthy controls. Moreover, the study aimed to assess the association of CAR with cognitive biases and psychopathological symptoms in people with personality disorders. Methods: A total of 43 individuals with personality disorders and 45 healthy controls were enrolled. Participants completed questionnaires measuring the severity of depressive symptoms, anxiety, cognitive biases, and psychotic-like experiences. Cortisol levels were measured in four morning saliva samples: immediately after awakening, and after 15, 30, and 45 min. Results: A significantly lower CAR was found among individuals with personality disorders, even after adjustment for age, sex, and the level of education. However, the receiver operating characteristic (ROC) curve analysis showed a relatively low area under the curve (AUC = 0.362). Furthermore, a significant negative correlation was observed between the CAR and the level of external attribution bias among individuals with personality disorders. No significant associations of the CAR with psychopathological symptoms and other cognitive biases were observed. Conclusions: Findings from this study indicate that the HPA axis activity might be altered in personality disorders. However, the clinical utility of this observation needs further studies in larger samples. External attribution might be related to the HPA axis alterations in this population.

## 1. Introduction

The hypothalamic–pituitary–adrenal (HPA) axis plays a key role in regulating the body’s response to stress. Activation of this axis leads to the release of cortisol. Cortisol secretion occurs in a cyclic manner throughout the day, reaching its highest concentration in the morning to gradually decrease thereafter. In addition, there is a sharp increase in cortisol levels within 30–45 min after waking, a phenomenon called the cortisol awakening response (CAR) [1]. The role of CAR is not fully understood. One major theory suggests that CAR enables the body to adapt to the demands of the day ahead [2]. It assists the body’s adaptive responses by facilitating the transition from sleep to wakefulness. As such, the magnitude of CAR may reflect an individual’s ability to cope with future challenges [2]. Furthermore, the CAR is associated with the reactivation of information from memory. The impairment of the typical CAR pattern may occur in the context of various psychosocial determinants, mental disorders, and physical health impairments [3]. The functioning of the HPA axis is highly reactive immediately following birth; however, during childhood, it may become hyporeactive to stress, and by adolescence, the HPA axis markers reach levels comparable to those observed in adulthood [4]. When the activation of this neuroendocrine system persists for an extended duration, it may exceed individual adaptive capacities, thereby increasing the risk of various disease outcomes, including stress-related mental disorders [5].

The dysregulation of the HPA axis has been associated with various mental disorders, such as depression and schizophrenia [6]. To date, few studies have been conducted on the relationship between personality disorders and CAR. Individuals with personality disorders, especially type B (histrionic, narcissistic, dissocial, and borderline disorder), show low resilience to situational stress, which is often due to traumatic childhood experiences [7]. Psychosocial stress can increase the risk of psychotic-like experiences [8], which is why people with personality disorders are considered a group at increased risk of psychosis [9,10]. Neuropsychological deficits and cognitive biases can contribute to the development and maintenance of hallucinations and delusions [11]. Research results suggest that people with personality disorders show cognitive biases similar to those observed in psychosis. Cognitive biases refer to the tendency of the human mind to interpret information in a way that may be incorrect or inaccurate. They include, e.g., jumping to conclusions, belief inflexibility, selective attention for threat, and external attribution bias [12].

In addition, individuals with personality disorders are particularly prone to develop problems with emotion regulation and show impulsive behavior. The dysregulation of emotions and impulses, instability of self-identity, as well as difficulties in interpersonal relationships often accompany self-aggressive and suicidal behavior [13]. As a result, personality disorders are often accompanied by depressive and anxiety disorders. Understanding the underlying psychological mechanisms (e.g., cognitive, emotional, and behavioral factors) that underlie these disorders is crucial to the development of effective psychotherapeutic interventions. The analysis of cognitive biases may not only help to understand potential mediators between personality disorders and other mental disorders, but also analyze some of the faulty affective and behavioral patterns [12,14].

To our knowledge, none of previous studies have investigated the interrelationships between cognitive biases, psychopathological symptoms, and CAR among individuals with personality disorders. Therefore, the aims of the present study were twofold. First, we aimed to compare the CAR between individuals with personality disorders and healthy controls. Next, we aimed to assess to whether CAR is associated with psychopathological symptoms and cognitive biases in people with personality disorders.

## 2. Materials and Methods

### 2.1. Participants

Participants were recruited between August and November 2023. The study group consisted of patients with a diagnosis of a personality disorder. The diagnosis of a specific subtype of personality disorder was assessed using the Structured Clinical Interview for DSM-5 Personality Disorders (SCID-5-PD) [15]. The comorbidity of other psychiatric disorders among the participants was verified using the Mini-International Neuropsychiatric Interview (M.I.N.I.) [16]. Individuals with personality disorders were enrolled from the Day Care Unit for Patients with Personality and Anxiety Disorders (Department of Psychiatry, Wroclaw Medical University, Wroclaw, Poland). They were recruited as a convenience sample before or at early stages of a group psychotherapeutic intervention delivered in this unit. In turn, healthy controls were recruited from the local community through advertisements. The following exclusion criteria were used in both groups: (a) the current and lifetime DSM-5 diagnosis of psychotic disorders; (b) chronic steroid therapy; (c) the current and lifetime diagnosis of endocrine diseases; (d) unstable comorbid somatic diseases; and (e) chronic use of anti-inflammatory drugs. All individuals gave a written consent to participate in the study. The protocol of this study was approved by the Bioethics Committee at Wroclaw Medical University, Poland (266/2023).

### 2.2. Measures

#### 2.2.1. Socio-Demographic Characteristics

In the case of all participants, the following socio-demographic characteristics were reported: age, sex, and the level of education.

#### 2.2.2. The Davos Assessment of the Cognitive Biases Scale (DACOBS)

The DACOBS is a self-reported questionnaire developed to measure a variety of cognitive biases associated with psychotic disorders including jumping to conclusions, belief inflexibility, selective attention for threat, external attribution bias, social cognition problems, subjective cognitive problems, and safety behaviors [11]. It consists of 42 questions, with responses rated on a 7-point scale. In our study, we used the Polish version of DACOBS validated and used by studies carried out in Poland [17,18,19]. The Cronbach’s alpha of the DACOBS was 0.902 in the present study.

#### 2.2.3. The Prodromal Questionnaire-16 (PQ-16)

The PQ-16 has been developed to screen for psychosis risk states [20]. It includes 16 items with two corresponding subscales. The first subscale refers to the presence of specific psychotic experiences (yes-or-no responses), while the second one measures associated distress (4-point responses). In our study, we focused on the presence subscale and used the Polish version of PQ-16, validated previously [21]. In our study, the Cronbach’s alpha for this subscale was 0.831

#### 2.2.4. The Patient Health Questionnaire-9 (PHQ-9)

The PHQ-9 measures the level of depressive symptoms over 2 preceding weeks. It consists of 9 items with potential responses on a 4-point scale. The total PHQ-9 score ranges between 0 and 27. Higher scores suggest a greater level of depressive symptoms [22]. In our study, we used the Polish version of PHQ-9 [23]. The Cronbach’s alpha of the PHQ-9 was 0.925 in the present study.

#### 2.2.5. The Generalized Anxiety Disorder-7 (GAD-7)

The GAD-7 measures the level of generalized anxiety symptoms across 2 preceding weeks [22]. It consists of 7 items with potential responses on a 4-point scale (0—not at all, 1—a few days, 2—more than half the time, 3—almost always). The total GAD-7 score is between 0 and 21, where higher scores suggest a greater level of generalized anxiety symptoms. In our study, we used the Polish version of GAD-7 [24]. The Cronbach’s alpha of the GAD-7 was 0.906 in the present study.

#### 2.2.6. The Level of Personality Functioning Scale Brief Form 2.0 (LPFS BF 2.0)

LPFS BF 2.0 is a self-reported questionnaire consisting of 12 items to evaluate the level of personality functioning as described in Section III of the DSM-5. Items are rated on a 4-point Likert scale ranging from 1 (completely false or very rarely true) to 4 (completely true or very often true) [25]. The total LPFS BF 2.0 score ranges between 12 and 48, where higher scores indicate a greater functional impairment. In our study, we used the Polish version of LPFS BF 2.0, which has been validated previously [26,27]. The Cronbach’s alpha of the LPFS BF 2.0 was 0.891 in the present study.

### 2.3. Measurement of Cortisol Levels

Respondents were given the cotton swabs (Salivette, Sarstedt, The Netherlands) to collect saliva. They were advised to refrain from consuming food and beverages upon awakening and during the measurement period. Samples were collected upon awakening and at 15, 30, and 45 min after awakening. Participants were instructed to store saliva samples in the home fridge upon their collection. The maximum time from collection to the delivery of samples to the laboratory was lower than 24 h. The electrochemiluminescence method (the Cobas Pro e801, Roche) and the CORTISOL II Elecsys Reagent for Roche (reference number: 06687733190) were used to perform measurements of salivary cortisol levels. These analyses were carried out by a commercial laboratory (Synevo, Warsaw, Poland).

### 2.4. Data Analysis

The χ^2^ test (categorical variables) and the Mann–Whitney U tests or *t* tests (continuous variables, depending on data distribution assessed using the Kolmogorov–Smirnov test) were used for the comparison of general characteristics across both subgroups of participants. The analysis of covariance (ANCOVA) was performed to test the association of group status (individuals with personality disorders vs. healthy controls) with CAR. Covariates included age, sex, and the level of education. The CAR was estimated using the area under the curve with respect to increase (AUCi) [28]. The Spearman rank correlation coefficients were computed to assess bivariate correlations. To further investigate if the CAR might show clinical utility in differentiating individuals with personality disorders, the receiver operating characteristic (ROC) curve analysis was carried out. The area under the curve (AUC) of >0.80 was interpreted as showing potential clinical utility [29]. Results of data analysis were interpreted as statistically significant if the *p*-value was <0.05. Statistical analyses were carried out in the SPSS software, version 28. Post hoc power analyses were performed using the G*Power software, version 3.1.9.7 [30].

## 3. Results

A total of 43 individuals with personality disorders and 45 healthy controls were recruited (Table 1). The majority of individuals with personality disorders met the criteria for a diagnosis of mixed personality disorder (60.5%). Both groups did not differ significantly with respect to age, sex, and the level of education. As expected, the level of personality functioning was significantly lower among individuals with personality disorders. Also, the level of all psychopathological symptoms and cognitive biases (except for jumping to conclusions) was significantly higher in individuals with personality disorders. No clinically relevant psychotic symptoms were observed.

The CAR was significantly lower among individuals with personality disorders (power = 0.722, Table 1) and showed normal distribution in the whole sample (D = 0.081, *p* = 0.200). This difference remained significant (F = 6.550, *p* = 0.012, power = 0.715) after covarying for age (F < 0.001, *p* = 0.992, power = 0.050), sex (F = 10.154, *p* = 0.002, power = 0.883), and education level (F = 2.490, *p* = 0.118, power = 0.345). However, the CAR was not found to be useful in differentiating individuals with personality disorders and healthy controls (AUC = 0.362, Figure 1). Bivariate correlations of CAR with psychopathological symptoms, cognitive biases, and the level of personality functioning are reported in Table 2. There was a significant negative correlation between CAR and the level of external attribution bias among individuals with personality disorders (Figure 2). This correlation was not significant in healthy controls. Also, other correlations appeared to not be significant.

## 4. Discussion

The present study provided insights into the differences in CAR between people with personality disorders and healthy controls, as well as their associations with psychopathological symptoms and cognitive biases. The results showed that people with personality disorders have significantly lower CAR compared to the control group, suggesting the presence of dysregulated HPA axis responses in this population. However, the CAR was not found to show potential utility as a marker to differentiate individuals with personality disorders and healthy controls. Previous studies have also reported altered responses of the HPA axis in people with conduct disorder and personality disorders. For instance, it has been observed that conduct disorder is associated with a lower volume of cortisol secretion attributable to a lower volume of cortisol secreted 30 min after awakening [31]. Another study showed a high percentage of individuals with blunted CAR among those with anxiety and personality disorders admitted to day hospital [2]. However, it is necessary to note that personality disorders represent a heterogeneous diagnostic construct, and it is reasonable to assume that various personality traits might be differentially associated with CAR. Indeed, it has been demonstrated that some dimensions of antisociality (i.e., callous-unemotional or manipulative traits and intentional aggression or conduct) are related to higher CAR [32]. In line with this observation, higher levels of anger and aggressiveness have been associated with a higher CAR in females with borderline personality disorder [33]. The association of personality traits with CAR have also been observed for extraversion and positive affect. Indeed, higher levels of extraversion and positive affect have been correlated with blunted CAR [34]. However, negative findings in this field have also been reported [35].

Moreover, a negative correlation between CAR and external attribution bias was found among individuals with personality disorder from our study. External attribution bias can be defined as a tendency to attribute the causes of various events or behaviors to social or physical environments. It has been widely investigated in the context of psychotic disorders. However, there is also evidence that individuals with borderline personality disorder might show high levels of external attribution biases [12]. Hormonal responses to stress, including those related to the release of cortisol, are known to impact the functioning of the prefrontal cortex and various executive functions [36]. Exposure to stress may also make individuals prone to relying on automatic processes for decision making [37]. In line with these considerations, Kubota et al. found that exposure to physiological stressors might increase the level of cortisol together with dispositional attributions of common daily behaviors and negative appraisals [38]. It has also been reported that the administration of hydrocortisone might shift cognitive processes towards error-prone intuitive thinking [39].

The heterogeneity of mental disorders leads to the identification of numerous biomarkers that may reflect various pathophysiological processes affecting etiology, diagnosis, and treatment [40]. Dysfunctions of the HPA axis may play important roles in personality disorders, allowing for a more precise matching of therapy to the individual needs of patients [2]. Indeed, it has been reported that CAR might moderate the efficacy of therapeutic interventions in patients with agoraphobia and panic disorder [41], as well as those with post-traumatic stress disorder [42]. In turn, the impact of psychotherapeutic interventions on CAR might be limited. Previous studies have not confirmed significant differences in CAR before and after psychotherapy [43,44]. However, there are reports on the impact of antidepressants on cortisol levels and potentially on CAR [45]. These studies suggest that the use of antidepressants contributes to a reduction in CAR [46]. Finally, it is necessary to note that blunted CAR might be associated with the presence of metabolic syndrome and its single components that are highly prevalent among individuals with various mental disorders [47,48].

## 5. Limitations

The present study has certain limitations. First, cortisol samples were collected on only one day, and objective measures to monitor the CAR protocol compliance were not used. Our sample was also relatively small, underpowered, and heterogeneous in terms of specific personality disorders. Diagnostic heterogeneity might potentially explain why CAR was not found to hold potential clinical utility as the measure differentiating personality disorders and healthy controls in the present sample. Moreover, the majority of participants were females, and thus the generalizability of findings might be limited. Another limitation is related to the fact that we did not control for potential confounding effects of treatments used by individuals with personality disorders from our sample. Furthermore, participants of the present study were enrolled as a convenience sample, and thus the generalizability of findings might be limited. Also, immune-inflammatory markers were not measured. Their assessment would provide further insights into the dysregulation of the HPA axis among individuals with personality disorders. Finally, a cross-sectional design does not allow information about causality.

## 6. Conclusions

Findings from this study indicate that individuals with personality disorders might show blunted CAR. This phenomenon might be associated with external attribution biases that are likely to occur in people with personality disorders. The findings indicate that aberrant cortisol responses among individuals with personality disorders might be related to biased cognitive processes. However, longitudinal and experimental studies are needed to unravel causality

## Figures and Tables

**Figure 1 brainsci-14-01040-f001:**
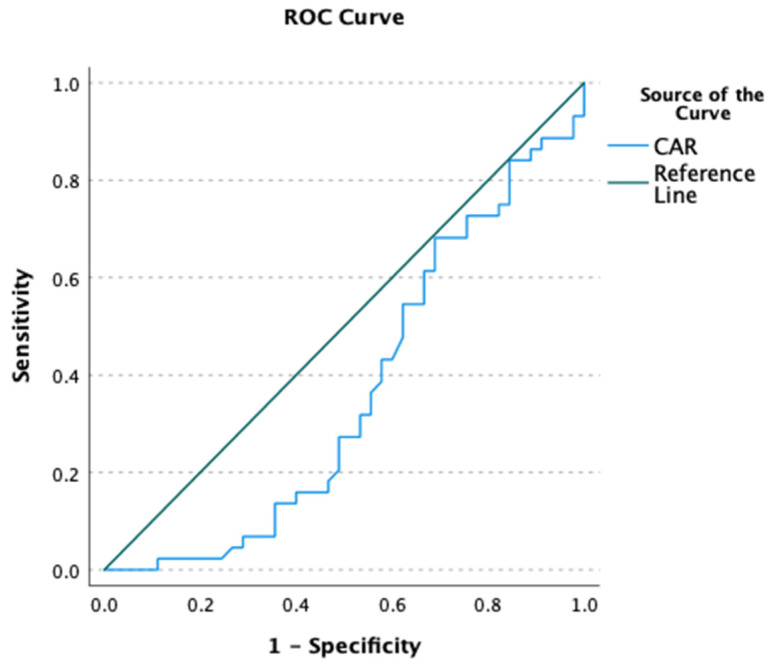
The ROC curve for cortisol awakening response (CAR).

**Figure 2 brainsci-14-01040-f002:**
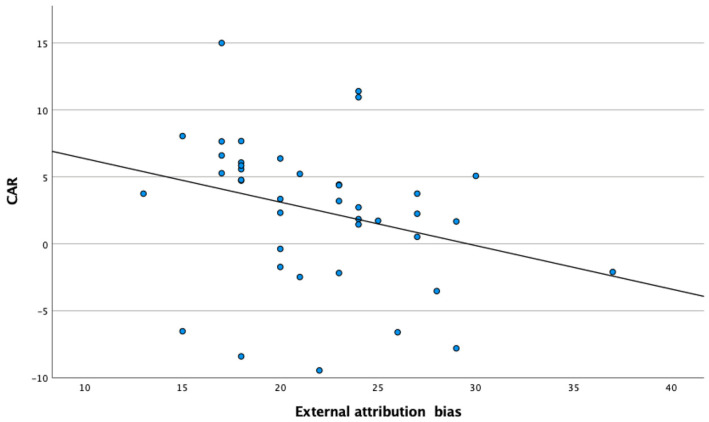
The correlation of cortisol awakening response (CAR) with the level of external attribution bias among individuals with personality disorders.

**Table 1 brainsci-14-01040-t001:** General characteristics of the sample.

	PD, *n* = 43	HCs, *n* = 45	Cohen’s d	*p*
Age, years	31.9 ± 10.2	28.7 ± 6.0	0.38	0.052
Gender, females	33 (76.7)	30 (66.7)	–	0.295
Education, higher	24 (55.8)	30 (66.7)	–	0.242
PQ-16, presence subscale	5.9 ± 3.6	2.7 ± 2.8	3.08	<0.001
PQ-16, distress subscale	9.9 ± 5.8	2.3 ± 2.4	4.40	<0.001
PHQ-9	17.0 ± 6.1	5.0 ± 4.5	5.31	<0.001
GAD-7	11.9 ± 5.5	4.7 ± 3.1	4.40	<0.001
DACOBS, safety behavior	14.9 ± 7.5	9.8 ± 4.1	5.99	<0.001
DACOBS, social cognition problems	26.8 ± 7.8	18.7 ± 6.7	7.25 *	0.002
DACOBS, jumping to conclusions	23.9 ± 5.8	25.9 ± 4.8	5.34 *	0.062
DACOBS, belief inflexibility	19.1 ± 4.0	14.8 ± 3.7	3.85	<0.001
DACOBS, attention for threat	23.9 ± 5.5	18.4 ± 5.4	5.45 *	<0.001
DACOBS, external attribution	21.9 ± 4.9	16.2 ± 4.3	4.59 *	<0.001
DACOBS, subjective cognitive problems	26.8 ± 7.8	18.7 ± 6.7	7.25 *	<0.001
LPFS BF 2.0	30.4 ± 6.1	19.6 ± 6.3	6.17 *	<0.001
CAR	2.4 ± 5.3	5.8 ± 7.1	0.55 *	0.025
Awakening cortisol levels	0.48 ± 0.21	0.45 ± 0.17	0.19	0.706
Mixed PD	26 (60.5)	–	–	–
Borderline PD	14 (32.6)	–	–	–
Obsessive compulsive PD	1 (2.3)	–	–	–
Histrionic PD	1 (2.3)	–	–	–
Dependent PD	1 (2.3)	–	–	–

* Differences assessed using *t*-tests (other comparisons were carried out using the Mann–Whitney U test). Note: CAR, cortisol awakening response; DACOBS, the Davos Assessment of Cognitive Biases Scale; GAD-7, the Generalized Anxiety Disorder-7; HCs, healthy controls; LPFS, the Level of Personality Functioning Scale; PD, personality disorders.

**Table 2 brainsci-14-01040-t002:** Bivariate correlations of CAR with the measures of psychopathology, cognitive biases, and the level of functioning.

	PD	HCs
PQ-16	*r* = −0.303, *p* = 0.057, power = 0.516	*r* = −0.056, *p* = 0.723, power = 0.065
PHQ-9	*r* = 0.087, *p* = 0.594, power = 0.086	*r* = −0.133, *p* = 0.395, power = 0.140
GAD-7	*r* = 0.257, *p* = 0.109, power = 0.389	*R* = −0.018, *p* = 0.907, power = 0.051
DACOBS, safety behavior	*r* = 0.086, *p* = 0.592, power = 0.085	*r* = 0.003, *p* = 0.986, power = 0.050
DACOBS, social cognition problems	*r* = 0.186, *p* = 0.244, power = 0.224	*R* = −0.170, *p* = 0.276, power = 0.201
DACOBS, jumping to conclusions	*r* = −0.123, *p* = 0.443, power = 0.123	*r* = −0.096, *p* = 0.542, power = 0.096
DACOBS, belief inflexibility	*r* = −0.103, *p* = 0.522, power = 0.100	*r* = −0.062, *p* = 0.693, power = 0.069
DACOBS, attention for threat	*r* = 0.123, *p* = 0.444, power = 0.123	*r* = −0.094, *p* = 0.549, power = 0.094
DACOBS, external attribution	*r* = −0.419, *p* = 0.006, power = 0.815	*r* = −0.072, *p* = 0.646, power = 0.075
DACOBS, subjective cognitive problem	*r* = 0.278, *p* = 0.079, power = 0.445	*R* = −0.031, *p* = 0.842, power = 0.055
LPFS BF 2.0	*r* = 0.176, *p* = 0.276, power = 0.205	*r* = −0.055, *p* = 0.727, power = 0.064

Note: CAR, cortisol awakening response; DACOBS, the Davos Assessment of Cognitive Biases Scale; GAD-7, the Generalized Anxiety Disorder-7; HCs, healthy controls; LPFS, the Level of Personality Functioning Scale; PD, personality disorders.

## Data Availability

The raw data supporting the conclusions of this article will be made available by the authors upon request.

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
