# Peer review of "Blunted Cortisol Awakening Response Is Associated with External Attribution Bias Among Individuals with Personality Disorders"

_brainsci, 2024, doi:10.3390/brainsci14101040_

Round 1
Reviewer 1 Report
Comments and Suggestions for Authors
The authors compare two groups, namely one group of patients with a personality disorder, and one group with healthy subjects. The found that the cortisol awakening response is blunted in the patients with the personality disorder. This finding should be discussed more in detail. The planning and performing of the study is good, and as well the statistical analysis. The conclusion should be explained more in detail. The English language is good enough, it should be overworked.
Comments on the Quality of English Language
The English language is good enough, it should be overworked.
Author Response
Dear Editor,
Dear Reviewers,
First of all, we would like to thank the Editor and all the reviewers for their detailed reviews and comments, and express our appreciation for the time and effort necessary to provide such insightful guidance. We would like to present the revised version of the manuscript entitled “Blunted cortisol awakening response is associated with external attribution bias among individuals with personality disorders”. We have carefully analyzed all comments. In this letter, we explain in detail on how we have revised the paper based on the provided comments and recommendations. All new parts of the manuscript are marked in orange color.
We hope the revisions have improved the manuscript in such a way that you will find it worthy of publication in the Brain Sciences.
On behalf of all authors,
Bogna Bogudzińska
Reviewer 2 Report
Comments and Suggestions for Authors
The study by Bogudzińska et al. examines the relationship between hypothalamic-pituitary-adrenal (HPA) axis activity, specifically the cortisol awakening response (CAR), and personality disorders. Below are suggestions to enhance the study's outcomes:
- Clarify the statement in the abstract: "Cortisol levels were measured in four morning saliva samples."
- Were participants screened for major psychiatric illnesses (e.g., schizophrenia, bipolar disorder) alongside personality disorders? Include this in the Methods, under Participants.
- Did the authors observe psychotic-prone symptoms in the personality disorder group? Please detail these findings in the Results section.
- The findings on CAR are intriguing, but it's unclear if the study assessed associations between CAR and inflammatory markers like CRP. Including such data would add value, especially regarding how a lower CAR may influence a pro-inflammatory response often linked to mental health disorders, particularly those involving psychoticism.
- Could the authors establish a CAR threshold predictive of personality disorders? This would have clinical relevance—consider including such data by developing a predictive model.
- Around line 194, discuss the potential therapeutic approaches (psychological or pharmacological) that could influence CAR and how they might impact outcomes for personality disorders, based on the study’s findings.
- Add a specific "Limitations" subsection following the Discussion.
Important suggestions for the Discussion section to further contextualize the findigs of the study:
- Discuss the importance of identifying biological markers for understanding the molecular basis of mental health (cite https://doi.org/10.1016/j.pneurobio.2011.05.006).
- Emphasize the role of CAR in metabolic syndrome, noting its links to insulin resistance, obesity, hypertension, and dyslipidemia, particularly in psychiatric populations (cite https://doi.org/10.1016/j.metabol.2009.10.024 andhttps://10.1021/acs.jproteome.2c00847).
- Address the relevance of novel biomarkers, such as extracellular vesicles, in understanding the molecular basis of psychiatric disorders (cite https://doi.org/10.1093/schbul/sby127, https://doi.org/10.7150/ijbs.79666, and https://10.3390/biomedicines12010129).
Author Response
Dear Editor,
Dear Reviewers,
First of all, we would like to thank the Editor and all the reviewers for their detailed reviews and comments, and express our appreciation for the time and effort necessary to provide such insightful guidance. We would like to present the revised version of the manuscript entitled “Blunted cortisol awakening response is associated with external attribution bias among individuals with personality disorders”. We have carefully analyzed all comments. In this letter, we explain in detail on how we have revised the paper based on the provided comments and recommendations. All new parts of the manuscript are marked in orange color. We hope the revisions have improved the manuscript in such a way that you will find it worthy of publication in the Brain Sciences.
On behalf of all authors,
Bogna Bogudzińska
REPLY TO COMMENTS OF THE REVIEWERS AND EDITOR
Reviewer 2
---------------------
- Clarify the statement in the abstract: "Cortisol levels were measured in four morning saliva samples."
We have added the following information to the abstract: “Cortisol levels were measured in four morning saliva samples: immediately after awakening, and after 15, 30, and 45 minutes”.
- Were participants screened for major psychiatric illnesses (e.g., schizophrenia, bipolar disorder) alongside personality disorders? Include this in the Methods, under Participants.
We have clarified that the Mini-International Neuropsychiatric Interview (M.I.N.I.) was used to assess comorbid mental disorders.
- Did the authors observe psychotic-prone symptoms in the personality disorder group? Please detail these findings in the Results section
No, clinically relevant psychotic symptoms were not observed in this sample. This has been clarified in the results.
- The findings on CAR are intriguing, but it’s unclear if the study assessed associations between CAR and inflammatory markers like CRP. Including such data would add value, especially regarding how a lower CAR may influence a pro-inflammatory response often linked to mental health disorders, particularly those involving psychoticism.
Unfortunately, immune-inflammatory markers were not assessed in the present study. We have pointed this out in the limitations.
- Could the authors establish a CAR threshold predictive of personality disorders? This would have clinical relevance—consider including such data by developing a predictive model.
Thank you for this remark. We have carried out the analysis of receiver operating characteristic (ROC) curve. However, the AUC was found to be low (0.362) indicating that the CAR was not found to be the measure differentiating individuals with personality disorders and healthy controls from the present study. Therefore, we did not perform further analyses to indicate a specific threshold. This observation has been reported and discussed in the manuscript.
6. Around line 194, discuss the potential therapeutic approaches (psychological or pharmacological) that could influence CAR and how they might impact outcomes for personality disorders, based on the study’s findings.
Important suggestions for the Discussion section to further contextualize the findings of the study:
Discuss the importance of identifying biological markers for understanding the molecular basis of mental health (cite https://doi.org/10.1016/j.pneurobio.2011.05.006).
Emphasize the role of CAR in metabolic syndrome, noting its links to insulin resistance, obesity, hypertension, and dyslipidemia, particularly in psychiatric populations (cite https://doi.org/10.1016/j.metabol.2009.10.024 and https://10.1021/acs.jproteome.2c00847).
Thank you for this crucial remark. We have added these references to the discussion section.
7. Add a specific "Limitations" subsection following the Discussion.
We have added the limitations section.

Reviewer 3 Report
Comments and Suggestions for Authors
Dear Authors
Thank you for allowing me to review this manuscript. I am going to make some comments with the aim of improving the manuscript. I want to be honest with you: I am not an expert in this subject. The comments I am going to make will be mainly focused on methodological aspects.
-The introduction is correct, although it is perhaps too synthetic. I think there is room for improvement, especially in terms of bibliographical support. There are only 6 references and 2 of them are more than 10 years old. Without being an expert on the subject, I understand that a lot is published in this area.
-Materials and Methods
-The study design should be indicated.
-For me the most important aspect to clarify in this research is how the study groups were formed. More information should be provided about the context in which the group with disorders and the control group were recruited. For example, the type of sampling and who carried out the recruitment should be explained. Another equally important aspect is the determination of the sample size. In principle you have two objectives, so either you have made a sample calculation based on the difference in the standard deviation to be detected in the scores of the different instruments, or based on the estimated correlation. Given that you have measured several outcome measures I understand that the sample calculation can be complex, but there must be a sample estimate that helps to assess the validity and power of the study
Indicate which socio-demographic variables you have measured. You give a lot of information about the instruments you have used to measure the different psychological aspects, and that is very good. But you should provide information on whether these instruments have been validated in previous studies in your context-country. As they already provide the internal consistency data obtained in their study, I don't think it is necessary for them to provide data such as the cronbach scores of the validation studies.
-The PQ-16 instrument indicates how the items score, but not the total scores for the two subscales.
In data analysis you have used non-parametric tests (Mann Whitney-Sperman). Given the sample size this may be appropriate, but were normality tests done? You could have calculated effect sizes for bivariate inferences (with cohen's d for example). If you are able to do so this would improve the manuscript.
-Results:
There is margin for improvement in some table. You could indicate with which statistical test each p-value has been obtained in table 1.
-In the discussion, it is a little bit like in the discussion. Perhaps the bibliographical support could be improved. Also, new lines of research based on these findings should be more clearly stated. While waiting for an answer regarding the sample calculation and the type of sampling, these aspects may need to be included in the limitations.
Author Response
Dear Editor,
Dear Reviewers,
First of all, we would like to thank the Editor and all the reviewers for their detailed reviews and comments, and express our appreciation for the time and effort necessary to provide such insightful guidance. We would like to present the revised version of the manuscript entitled “Blunted cortisol awakening response is associated with external attribution bias among individuals with personality disorders”. We have carefully analyzed all comments. In this letter, we explain in detail on how we have revised the paper based on the provided comments and recommendations. All new parts of the manuscript are marked in orange color.
We hope the revisions have improved the manuscript in such a way that you will find it worthy of publication in the Brain Sciences.
On behalf of all authors,
Bogna Bogudzińska
- The introduction is correct, although it is perhaps too synthetic. I think there is room for improvement, especially in terms of bibliographical support. There are only 6 references and 2 of them are more than 10 years old. Without being an expert on the subject, I understand that a lot is published in this area.
We have revised the introduction by referring to more recent studies.
- For me the most important aspect to clarify in this research is how the study groups were formed. More information should be provided about the context in which the group with disorders and the control group were recruited. For example, the type of sampling and who carried out the recruitment should be explained. Another equally important aspect is the determination of the sample size. In principle you have two objectives, so either you have made a sample calculation based on the difference in the standard deviation to be detected in the scores of the different instruments, or based on the estimated correlation. Given that you have measured several outcome measures I understand that the sample calculation can be complex, but there must be a sample estimate that helps to assess the validity and power of the study
Thank you for this crucial remark. More information about recruitment procedures has been added to the manuscript. Moreover, post-hoc power calculations have been performed and are reported in the results section as well as Table 2. In the majority of tests, the power was lower than 80% and this point has been addressed in the limitations.
- Indicate which socio-demographic variables you have measured. You give a lot of information about the instruments you have used to measure the different psychological aspects, and that is very good. But you should provide information on whether these instruments have been validated in previous studies in your context-country. As they already provide the internal consistency data obtained in their study, I don't think it is necessary for them to provide data such as the cronbach scores of the validation studies
Age, sex, and the level of education were recorded. This information has been added to the measures subsection. Moreover, we have revised the information about measures used in the present study as requested by the Reviewer.
- The PQ-16 instrument indicates how the items score, but not the total scores for the two subscales.
This information has been added to Table 1.
- In data analysis you have used non-parametric tests (Mann Whitney-Sperman). Given the sample size this may be appropriate, but were normality tests done? You could have calculated effect sizes for bivariate inferences (with cohen's d for example). If you are able to do so this would improve the manuscript.
We have clarified that either the Mann-Whitney U test or t-tests were used (depending on data distribution). The CAR showed normal distribution and thus we did not proceed to any data transformation. We have also added the Cohen’s d for all continuous variables tested for between-group differences.
- There is margin for improvement in some table. You could indicate with which statistical test each p-value has been obtained in table 1.
We have added this information to table 1.
- In the discussion, it is a little bit like in the discussion. Perhaps the bibliographical support could be improved. Also, new lines of research based on these findings should be more clearly stated. While waiting for an answer regarding the sample calculation and the type of sampling, these aspects may need to be included in the limitations.
We have revised the discussion section and added more points to the limitations.

Round 2
Reviewer 2 Report
Comments and Suggestions for Authors
The authors have successfully addressed all my suggestions.
Comments on the Quality of English Language
-